# Suicide Overall and Suicide by Pesticide Rates among South Korean Workers: A 15-Year Population-Based Study

**DOI:** 10.3390/ijerph16234866

**Published:** 2019-12-03

**Authors:** Yangwoo Kim, Jeehee Min, Soo-Jin Lee

**Affiliations:** 1Department of Occupational and Environmental Medicine, Hanyang University Seoul Hospital: Seongdong-gu, Seoul 04763, Korea; oem.ywkim@gmail.com (Y.K.); maryann87@naver.com (J.M.); 2Department of Public Health Science, Graduate School of Public Health, and Institute of Health and Environment, Seoul National University, Gwanak-gu, Seoul 08826, Korea; 3Department of Occupational and Environmental Medicine, College of Medicine, Hanyang University, Seongdong-gu, Seoul 04763, Korea

**Keywords:** suicide rate, suicide by pesticide, occupational group, suicide prevention, South Korea, population-based study

## Abstract

Suicide is a major public health concern in South Korea, and self-poisoning by pesticides is one of the common methods of suicide. Pesticide ban policies have been successful for suicide prevention; however, no studies have shown their effect according to occupational groups. The present study analyzed suicide and suicide by pesticide rates among South Korean workers aged 15–64 in 2003–2017, their associations with occupational groups, and the impact of three major economic indices on these factors. Workers in the agriculture, forestry, and fishery industries had relative risks of 5.62 (95% CI: 5.54–5.69) for suicide overall and 25.49 (95% CI: 24.46–26.57) for suicide by pesticide. The real gross domestic product had a positive association with suicide overall only in the last five-year period investigated in this study, and the unemployment rate consistently had a positive association. The economic status and policy for suicide prevention affected suicide and suicide by pesticide rates differently among occupational groups and different time periods. Policy addressing suicidal risk for different occupational groups should be of concern in South Korea.

## 1. Introduction

The suicide rate of South Korea has increased in the past decades. It was 148 per 1,000,000 in 2000 and increased to 283 per 1,000,000 in 2015 [1]. It is a major public health concern in South Korea, so many kinds of political interventions for suicide prevention have been introduced. Mainly, they focused on the youth and the elderly by providing education and welfare [2,3,4,5]. Restricting access to lethal methods of suicide was one of those interventions and it was effective for prevention [6].

Self-poisoning by highly toxic pesticides is a common suicidal method, so political intervention has been actively adopted in agricultural countries, including Sri Lanka and Bangladesh, and also in high-income countries, including the UK and South Korea [7,8,9,10,11,12]. South Korea introduced a complete ban on highly toxic pesticides in November 2012, and it effectively decreased both the suicide overall and suicide by pesticide rates [7,13,14,15]. The World Health Organization confirmed that South Korea made one of the most successful policies for suicide prevention [16].

Furthermore, sociodemographic factors like age, sex, and occupation affect the suicide rate. Men are more vulnerable than women, and older people commit more suicides than younger people; that is, the male suicide rate of South Korea (371 per 1,000,000) was more than two-fold that of the female rate (177 per 1,000,000), and that of people over the age of 70 was 1160 per 1,000,000 in 2012.

Additionally, suicide rates are different based on employment status and occupational groups [17,18]. People with occupations show lower suicide ideation and suicide rates than people without occupations, and specific occupational groups, such as agriculture, forestry, and fishery, have higher suicidal rates than other groups [19,20,21,22]. There is a social discrepancy in occupational groups, which makes suicide rates different [17,19,22].

Besides personal variables, societal circumstances are also crucial factors. The three major economic indicators, which are the real gross domestic product (RGDP), the unemployment rate (UnR), and the consumer price index (CPI), represent the status of a country. They have an association with suicidal ideation, suicidal attempts, and suicides of South Korea and Western countries [23,24,25,26]. In addition, the economic crises of the late 1990s in Asia and 2008 worldwide were inflection points in the surge of suicides [27,28].

However, there are no satisfactory results about the trends of suicide overall and suicide by pesticide rates and the effects of suicide preventions by occupational groups. It is reasonable to assume that policies for suicide prevention have different effects on each occupational group i.e. pesticides are primarily related to the agriculture, forestry, and fishery industries. In this study, we analyzed the association of ecological factors, including the time before and after the pesticide ban policy was introduced and the other factors mentioned above, with 15-year suicide rates. Finally, we showed that suicide overall and suicide by pesticide rates of workers in the agriculture, forestry, and fishery industries are higher than other occupational groups and even the general population, and suicide by pesticide decreased after the pesticide ban policy was introduced.

## 2. Materials and Methods

### 2.1. Data Sources

Data used in this study comprise three parts: (1) death records, (2) occupational populations, and (3) macroeconomic indicators. Death records and population were extracted from the microdata provided by the Korean National Statistical Office (KNSO) on July 15, 2019. The information of deaths from the National Death Records (NDR), which is the official record of deaths in South Korea, and the target years were the 15-year period from 2003 to 2017.

Cause of death is recorded by the International Classification of Diseases 10th edition (ICD-10) in the NDR. The ICD-10 codes for deaths by suicide are intentional self-harm (X60–X84), consequences of intentional self-harm (Y87.0), and suicide by pesticide (X68). The total number of suicides and suicides by pesticides by year is shown in Appendix A. The economic indicators were from the Korean Statistical Information Service (KOSIS). The RGDP is a percent of the rate of the nominal gross domestic product from one year to another, the UnR is a percent of the sum of people looking for a job in the last four weeks that were unsuccessful divided by the reference population, and the CPI is the average of the prices of selected main consumer goods relative to those in a reference year [23,29]. In the present study, we used the rate of the CPI from one year to another instead of the CPI itself because the CPI characteristically increases by year, and the rate is a better fit for time trend analysis.

### 2.2. Study Population

The study population was from the Economically Active Population Survey (EAPS), which is the national survey in South Korea for working population statistics such as the employment and unemployment rates. The target age of the EAPS is 15 to 64, which is the working-age declared by the International Labour Organization (ILO). This study set the target age as 15 to 64 for fully utilizing the EAPS results. The ILO defines a worker as a person who worked for longer than an hour in the past week. There are some differences in the definition of workers by the KNSO and the ILO. The KNSO defined a worker in the same way as the ILO until 2012; however, since then it changed the definition to mean a person who worked for longer than an hour in the past four weeks. In the present study, the ILO definition was adopted to calculate consistent estimations from the 15-year data. The population of each occupational group is shown in Appendix A.

Occupations were classified into five groups based on the Korean Standard Occupational Classification 6th edition; they are (1) managers and professionals (MNP), (2) officers and workers in services and trades (OST), (3) workers in agriculture, forestry, and fishery (AFF), (4) skilled manual labor workers (SKL), and (5) unskilled manual labor workers (USL).

The reference population used for statistical analysis came from the Korea National Consensus Survey (KNCS), which includes workers and non-workers aged 15–64. The KNCS measured the population once every five years (2005, 2010, and 2015), so the 2005 population was used as a reference for 2003–2007, 2010 for 2008–2012, and 2015 for 2013–2017.

### 2.3. Statistical Analysis

The number of workers in South Korea by sex, age, and occupation was estimated using the microdata from the EAPS. The EAPS from sample households were conducted each month, so the total number could be estimated by adjusting weights. Sociodemographic characteristics of suicide and pesticide-use suicide data collected from the NDR were also recorded.

The present study calculated the crude mortality rate (CMR) as the number of deaths from the NDR divided by the population from the EAPS matched for their age, sex, and job and converted it to units per 1,000,000. This rate is not a cohort statistical result, but a hypothetical formula based on population-based studies. It resulted in the standardized mortality ratios (SMRs) for the observed deaths in a population for the specific sex, 5-year age category, and occupation groups divided by the expected values estimated by the deaths of the reference population in each sex and age group. By this calculation, it was possible to directly compare specific occupational groups and the general population. If the mortality rate was the same as the reference, the value of SMR would be 100.

The 95% confidence intervals (CIs) were calculated by the Vandenbroucke method, which provides the CIs through shortcuts. The Vandenbroucke method, which assumes that the deaths show a Poisson distribution, is suitable for dealing with a large number (>20) of deaths and creates simple approximations for comparison [30].

Regression analyses were performed to show associations between suicide rates and personal and social factors. The crude rates of the specific population were the outcome variables, which were expected to have a Poisson distribution, and the other factors were the predictor variables. This method was used to analyze the trend of occupational diseases in previous studies [31]. First, the logistic regression models for occupational groups stratified with age and sex were calculated. As a result, relative risks (RRs) were determined for each occupational group compared to SKL over the 15-year period death statistics. Moreover, the linear regression models were made for economic indicators by each year stratified with occupational groups. These models showed the effects of social factors on the suicide and suicide by pesticide rates by occupation and the time trend of the effects. All statistical and mathematical analyses were performed by the statistical software R, version 3.6.1 (2019-07-05) [32].

### 2.4. Ethics Statement

The present study was approved by the Institutional Review Board of Hanyang University (HYU-2019-06-013) and only data without personal identifiers from the KNSO were used.

## 3. Results

### 3.1. Standardized Mortality Ratios and Crude Mortality Rates of Suicide and Suicide by Pesticide

All CMRs and SMRs of the workers aged 15–64 for suicide overall and suicide by pesticide by year are summarized in Table 1. The time trends of CMRs and SMRs over the 15-year period are visualized in Figure 1 and Figure 2. CMRs for all working-age populations, including workers and non-workers, are listed in Appendix A, and shown in Appendix A. There was a dramatic decrease in the CMR of suicide overall from 2005 to 2006 (–16.5%), with the lowest point occurring in 2006 (129 per 1,000,000). However, it largely increased after three years, in 2009; the CMR shows its two highest peaks in 2009 and 2013 (191 and 194 per 1,000,000). After 2013, it decreased slightly to 161 per 1,000,000 in 2017.

The SMR of suicide overall, which is a representative marker comparing workers to the general population, showed a different pattern. It started with 65.4 in 2003, directed downward until 2007 at 53.7 (95% CI: 51.8–55.7), and flattened from 2008 at 55.3 (95% CI: 53.5–57.3) to 2012 at 58.8 (95% CI: 57.0–60.6). However, it surged in 2013 (+13.8%) and decreased between 2016 and 2017 (–4.2%).

The CMRs of suicide by pesticide diminished through the 15-year period. It achieved a decrease every year from 2003 to 2017 (–90.9%, 55 to 5 per 1,000,000). On the contrary, the SMRs of suicide by pesticide showed a complex fluctuation. The lowest point was in 2007 at 61.5 (95% CI: 57.0–66.2) and two peaks were seen in 2009 at 70.0 (95% CI: 64.8–75.4) and 2013 at 78.1 (95% CI: 69.6–87.2).

We can confirm that the suicide overall and suicide by pesticide rates of workers were different in age and sex, which was observed in the general population. The CMRs were higher for men and the elderly compared to women and the younger cohort (Appendix A).

However, the SMR of suicide overall by age group showed a peculiar trend. From 2003 to 2010, intervals between those aged 15–39 and those age 40–64 became close, and finally, the SMRs were similar in 2011. After 2013, the trend changed in that the SMR of the younger group was larger than that of the older until 2017.

### 3.2. The Time Trend and Relative Risks for Occupational Groups

The CMRs and SMRs of suicide overall and suicide by pesticide are tabulated by occupational groups through the 15-year period in Appendix A. For suicide overall, the SMRs and the CMRs did not change significantly with the OST and SKL groups (Figure 3). The AFF group always showed the highest values, and the SMRs were larger than 100 each year; the suicide rate of the AFF was worse than of the general population. The MNP groups slightly increased through the 15-year period, both in the SMRs and the CMRs; the SMR increased 274.0% from 18.0 (95% CI: 14.7–21.7) in 2003 to 67.4 (95% CI: 63.0–72.0) in 2017, and the CMR increased 237.0% from 46 (95% CI: 37.8–56.0) in 2003 to 155 (95% CI: 145.3–166.1) in 2017.

Furthermore, the USL group showed a large increment in both the SMR and CMR of suicide overall between 2011 and 2013: 47.5 (95% CI: 43.1–52.2) to 98.1 (95% CI: 90.8–105.7) in the SMR, and 152 to 257 per 1,000,000 in the CMR. In 2014, the SMR of the USL group reached over 100. For suicide by pesticide, the AFF workers had much higher suicide rates than other workers, and the CMR of AFF workers decreased over the 15-year period while the SMR fluctuated irregularly.

Table 2 summarizes the RRs for each occupational group by synthesizing all death data through the 15-year period (Table 2). Moreover, stratification was performed to confirm the effect on age and sex; the strata were younger male, older male, younger female, and older female, where the younger corresponds to people age 15–39 and the older corresponds to people age 40–64. The time trends of SMRs and CMRs by occupational groups over the 15-year period are shown in Figure 3.

The SKL group showed the lowest suicide rate, so it was set as the reference for analyses. For all occupational groups without stratification, the AFF group had the highest RR for suicide overall, 5.62 (95% CI: 5.54–5.69), followed by OST and USL. The MNP group showed the lowest value, 1.44 (95% CI: 1.42–1.47). The AFF also had the highest RR for suicide by pesticide, 25.49 (95% CI: 24.46–26.57), followed by the USL and OST groups; however, only the MNP group had a lower risk than SKL, 0.83 (95% CI: 0.78–0.88). The AFF group recorded the highest value of suicide and suicide by pesticide rates in all strata, and the younger male and younger female strata represented the MNP groups with the lowest RR for suicide and suicide by pesticide.

### 3.3. The Impacts of Suicide Prevention and Economic Indicators on Suicide and Suicide by Pesticide

First of all, we confirmed the effect of time flow by estimating RRs for the five-year periods of 2008–2012 and 2013–2017, with reference to the first five-year period of 2003–2007, without any variables. This estimation only showed the effect of time on suicide and suicide by pesticide in Table 3. In the third five-year period, it could be assumed that the effect of the pesticide ban policy for suicide prevention occurred. The suicide overall rate for all occupations did not change over the 15-year period, that is, the RR of 2008–2012 was 0.95 (95% CI: 0.94–0.96) and of 2013–2017 was 1.01 (95% CI: 1.00–1.02).

However, the suicide by pesticide rate apparently decreased; the RR for 2008–2012 was 0.60 (95% CI: 0.59–0.61) and for 2013–2017 was 0.22 (95% CI: 0.22–0.23). Additionally, the suicide overall rate of the MNP group showed an enormous increase with time; the RR for 2008–2012 was 2.18 (95% CI: 2.11–2.25) and for 2013–2017 was 2.66 (95% CI: 2.58–2.74). The SKL group also showed an increase.

Generalized linear regression models were used as indicators of the association between suicide and suicide by pesticide, and three major economic indicators, which were RGDP, UnR, and CPI, were considered with the effect of time. The time trends of these indices are shown in Appendix A. These models showed interactions with time per year, so regressions were divided by each five-year period, with the first as 2003–2007, the second 2008–2012, and the third 2013–2017. The SKL group had the lowest RR, and the AFF group had the highest RR for the models. The calculated β values are tabulated in Table 4, and an overview is presented in Figure 4 and Figure 5.

For the suicide overall rate of all occupational groups, the RGDP showed no significant effect on the first and second five-year periods (β_GDP_ = −0.03 and 0.00) but a positive association in the last five-year period (β_GDP_ = 0.34); the UnR represented positive associations for each five-year period (β_UnR_ = 0.15, 0.27, and 0.06), and the CPI had a positive association in the first five-year period (β_CPI_ = 0.16) but changed to a negative association in the last period (β_GDP_ = −0.27). These patterns were similarly repeated in the suicide overall rate of the AFF group. However, the SKL had a different trend; the effect of UnR was negative in the first period (β_GDP_ = −0.20) and not statistically significant in the other periods.

Macroeconomic indicators were associated with suicide by pesticide differently compared to suicide overall. The positive associations of RGDP became huge in the last five-year period (β_GDP_ = 1.26 for all occupations, β_GDP_ = 1.20 for SKL, and β_GDP_ = 1.29 for AFF). The UnR and CPI showed strong negative associations with suicide by pesticide only in the last five-year period.

## 4. Discussion

The present study investigated suicide among South Korean workers in 2003–2017. The CMR for suicide overall shows a considerable increase between 2008 and 2009 and a slight one between 2011 and 2013. However, in a previous study [4], the general population had a different pattern with an enormous increase between 2008 and 2009, followed by a continuous decrease. The suicide rate among the elderly aged over 70 diminished significantly due to active suicide preventions for them, such as phone-based interventions and programs by the Mental Health Welfare Center [3].

However, the suicide overall rate among the general population of men aged 30–40 increased in 2011 and 2013; there was no targeted policy for them [4,5]. Similarly, suicide prevention for workers was weaker than for the elderly, so this could result in the suicide rate of workers showing an increase, while that of the general population showed a decrease [33].

The SMR shows a direct comparison of the characteristics of workers in the working-age population with the general population at the same age; it is not an absolute value but a relative one, where the number of deaths in the general population is set to 100 [30]. In general, people with occupations show lower suicide rates than people without occupations.

The SMR of suicide overall went down from 2003 to 2007, decreasing from 65.4 (95% CI: 63.2–67.7) to 53.7 (95% CI: 51.8–55.7), and surged from 2011 to 2015, increasing from 54.6 (95% CI: 52.9–56.2) to 67.7 (95% CI: 65.6–69.7). Furthermore, the SMR of older people (40–64 years old) was larger than that of younger people (15–39 years old) until 2011; however, the younger had higher suicide rates than the older after 2014. The global economic crisis that occurred in 2008 might be associated with this phenomenon because the change started after 2008 [27]. Suicide preventions in South Korea focused on the elderly and unemployed, so workers would not be protected, which could explain the relative increase of the SMR of suicide overall among workers [2,3,4,5]. In regard to gender, men are more suicidal than women, even among workers in this study. These are unique trends not similar to any of those in the general population.

During the most recent 15 years, the CMR of suicide by pesticide among South Korean workers of the working-age population has continuously decreased. At first, the present study confirms that the pesticide ban policy led to decreases in suicide by pesticide in workers, as well as in the general population [7,13,14,15]. In a previous study, the age-standardized suicide by pesticide rate among the general population in South Korea went down from 2011 to 2013: 5.26 to 2.67 per 1,000,000; the complete regulation of highly toxic pesticides was introduced in 2012 [13,14]. This reduction was significant in men, the elderly, and rural residents. Additionally, pesticide ban policies have been successful in other countries; a meta-analysis proved that national bans of specific pesticides in six countries, including South Korea, showed consistent results concerning decreases in suicide by pesticide [7,9,12,13,14,34,35].

The present study also finds more considerable reductions in the CMR of suicide by pesticide among South Korean workers in men, aged 40–64, and after 2013 than female, aged 15–39, and before 2011. In contrast to the decrease in the CMR of suicide by pesticide among workers from 2011 to 2013, the SMR increased from 65.2 (95% CI: 60.1–70.6) to 78.1 (95% CI: 69.6–87.2). This distinction is due to more reductions of suicide by pesticide in the general population than in the working population.

Occupation is one of the social conditions that affects suicide rates [17,18,19,20,25,31]. Particular jobs, such as medical professionals, soldiers and veterans, and farmers, showed higher suicidal ideations, attempts, and rates than others [18]. These associations were attributed to job strain, stress, long working hours, and convenient access to suicidal methods [17].

One advantage of the present study is that the rates and ratios were calculated more accurately than previous studies by using the results of EAPS, employment and unemployment surveys of the working population taken monthly every year. An analysis by occupation groups in a previous study also found workers in the agricultural, fishery, and forestry industries had more significant suicide rates than workers in other occupational groups [19,22].

The present study confirms that the AFF group was more vulnerable to suicide and suicide by pesticide than others. Even the SMR of AFF was higher than 100 every year, which means that the AFF group showed higher suicidality than the general population; suicide by pesticide decreased, but the suicide overall rate remained dangerously high (488 per 1,000,000 in 2017). Pesticide ban policies achieved a more successful effect on suicide by pesticide of the AFF group than others; however, there was no such decrement in the suicide overall rate of AFF.

In addition, the MNP group annually increased both in the CMR and SMR for suicide overall, especially for men. The CMR of men in the MNP was 58.6 in 2003 and 244.7 in 2017, and the SMR was 16.9 in 2003 and 74.1 in 2017. These values were still lower than those of men in the OST or USL group, however, men in the MNP group became more vulnerable to suicide in the past decades (Appendix A). This phenomenon might be due to the change of population structure of occupations. The population of the MNP group rapidly increased after the economic crisis in 2008; men and women in this group in 2007 numbered 1,536,126 and 909,955, respectively, and in 2008 they numbered 2,892,005 and 1,881,010, respectively (Appendix A). The people in the SKL or USL groups who had lost their jobs could come to be the MNP group by establishing their own small businesses, so the MNP group would be more socially weak than in the past. This explanation is compatible with results of a positive association between the UnR and suicide rates of all occupational groups.

The USL group experienced a significant increment through 2011 to 2014, and the SMR of the USL group reached 100 in 2014. The analysis of macroeconomic indicators in the present study implies that UnR affected the increment in the suicide overall rate in 2011 to 2014 because it showed a more substantial value (β_UnR_ = 0.27) than RGDP and CPI in the second five-year period, 2008–2012.

For comparison between each occupational group, RRs for suicide and suicide by pesticide were calculated through the 15-year period by using SKL as the reference group with age and sex stratification. The AFF group showed the highest RR in both suicide overall and suicide by pesticide without stratification, followed by the OST, USL, and MNP groups. The MNP group was riskier than the SKL group for the elderly but less so for the young in suicide overall. The young female stratum showed relatively lower RRs than others, which means that the SKL group of the young female group had a higher suicidal risk than other occupations. For suicide by pesticide, the AFF group also had the highest RR among occupational groups in all strata; and the older female group showed the most significant RR for AFF and the older male group the lowest.

RRs were estimated for each five-year period using the first five-year period as the reference to confirm the effects of time and certain events; the global economic crisis occurred in 2008 and the complete ban of highly toxic pesticides in 2012 [27]. RRs for suicide by pesticide experienced dramatic decreases in the second period, and the values halved from the second period to the third period, for almost all occupational groups. This confirms the time trend of suicide by pesticide, and the downward trend became much more severe after 2013. The suicide overall rate for all workers did not change significantly; meanwhile, rates for the MNP group increased and those for the AFF group decreased with time.

The RGDP, UnR, and CPI are three major macroeconomic indicators of a country and are associated with suicide and suicidal behaviors [23,24,25,26,27,29,33]. The present study made regressions to their effects on the suicidality of occupational groups for each period. First, the RGDP did not significantly affect the first and second five-year periods. However, it had a positive association in the third period both in suicide overall and suicide by pesticide. The effects of the RGDP were not consistent in previous studies; there was a weak positive association in men and a weak negative association in women in the general population of Europe from 2001–2011, and the MNP and OST groups were the occupational groups with statistically significant positive associations in South Korea [19,24]. The results of the present study inferred that the effect of the RGDP on suicidality could depend on the time period, not the characteristics of the population.

The UnR showed consistently positive effects for the suicide overall rate without job stratification and with the stratum of workers in the AFF group. Meanwhile, the SKL group had a negative association with the UnR in the first five-year period and no significant association in other periods. For suicide by pesticide, the UnR had a negative association in the first and second five-year periods, but no significant association in the last five-year period. In a previous study, the UnR was related to the suicide rates only in low social class occupational groups such as unskilled labor and agriculture, forestry, and fishery [19,27,28]. The SKL group has a relatively higher socioeconomic status, so their suicide overall rate would not be sensitive. Suicide by pesticide, which is preferred as a suicidal method in lower-class occupations, showed a positive association with the UnR, but the relationship disappeared in the third five-year period after the number had mainly decreased [13,14].

The CPI was calculated as the percentage of the rate from one year to another. The means of the CPI are 2.92, 2.98, and 1.24 for each five-year period analyzed in this study. Suicide and suicide by pesticide had a positive association in the first period and a negative association in the third period. When the CPI increased, the household expenditure increased, and the lower socioeconomic class felt stronger negative effects. After 2013, the CPI fell below 1.0, so its effects on the lower class were smaller, which resulted in a negative association with suicidality.

There are some limitations to this study. First, causal inference is difficult to determine because it is not a cohort study nor a randomized clinical intervention; it is an ecological study that used population-based data. Ecological studies are not suitable for finding confounding variables [36,37]. The results of this study mean that suicidal rates of occupational groups have different characteristics; however, the specific occupational group is not a cause of suicide. It is the same for economic factors. Associations between suicide rates and economic factors only show trends in the long run, but it is not able to conclude that the change of macroeconomics causes suicide. Causal inferences by population-based studies might lead to an ecological fallacy.

Second, the data sources of population and death were different; the population data source was the EAPS, and that of death was the NDR. The EAPS and the NDR are reliable national databases, but they were collected separately. Therefore, the suicide rates calculated by these data were hypothetical. The exact rates should be from an observed population by a cohort design [38]. However, this study provided an overview of suicidalities among South Korean workers to help to design further studies based on a cohort or a big data set and policy interventions for the specific occupations. 

## 5. Conclusions

Age, sex, and time period effects and the impacts of macroeconomic indicators are different in suicide and suicide by pesticide by occupational groups. The AFF group showed higher suicide rates than the general population, and the suicide rates of the USL group rapidly increased in the most recent five-year period. Suicide by pesticide largely decreased in all occupational groups after 2012, when the pesticide ban policy was introduced, but the AFF group remained as the highest risk group. The population structure of occupational groups changed over time, and the effects of macroeconomic indicators on suicide rates also changed during each period. The RGDP had a positive association with suicide overall rate in the last five-year period, and the UnR also had a positive association, but it disappeared with suicide by pesticide in the third period. The results of this study show that suicidal prevention should be considered for risky occupations in South Korea.

## Figures and Tables

**Figure 1 ijerph-16-04866-f001:**
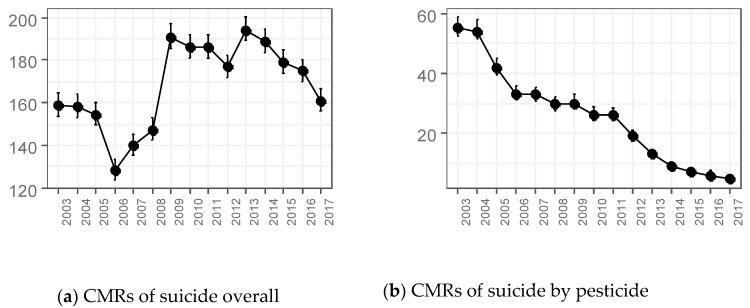
Time trends of crude mortality rates (CMRs) of suicide and suicide by pesticide among workers aged 15–64. (**a**) CMRs of suicide overall. (**b**) CMRs of suicide by pesticide.

**Figure 2 ijerph-16-04866-f002:**
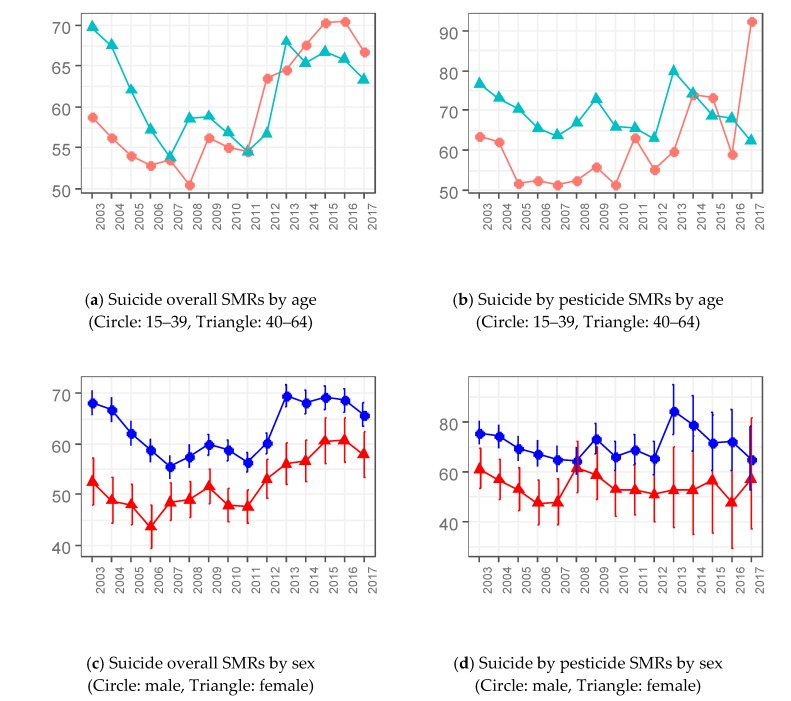
Time trends of standardized mortality ratios (SMRs) of suicide and suicide by pesticide among workers age 15–64. Figures in the left column deal with suicide overall and those on the right column with suicide by pesticide. (**a**) Suicide overall SMRs by age (Circle: 15–39, Triangle: 40–64). (**b**) Suicide overall SMRs by sex (Circle: male, Triangle: female). (**c**) Suicide by pesticide SMRs by age (Circle: 15–39, Triangle: 40–64). (**d**) Suicide by pesticide SMRs by sex (Circle: male, Triangle: female).

**Figure 3 ijerph-16-04866-f003:**
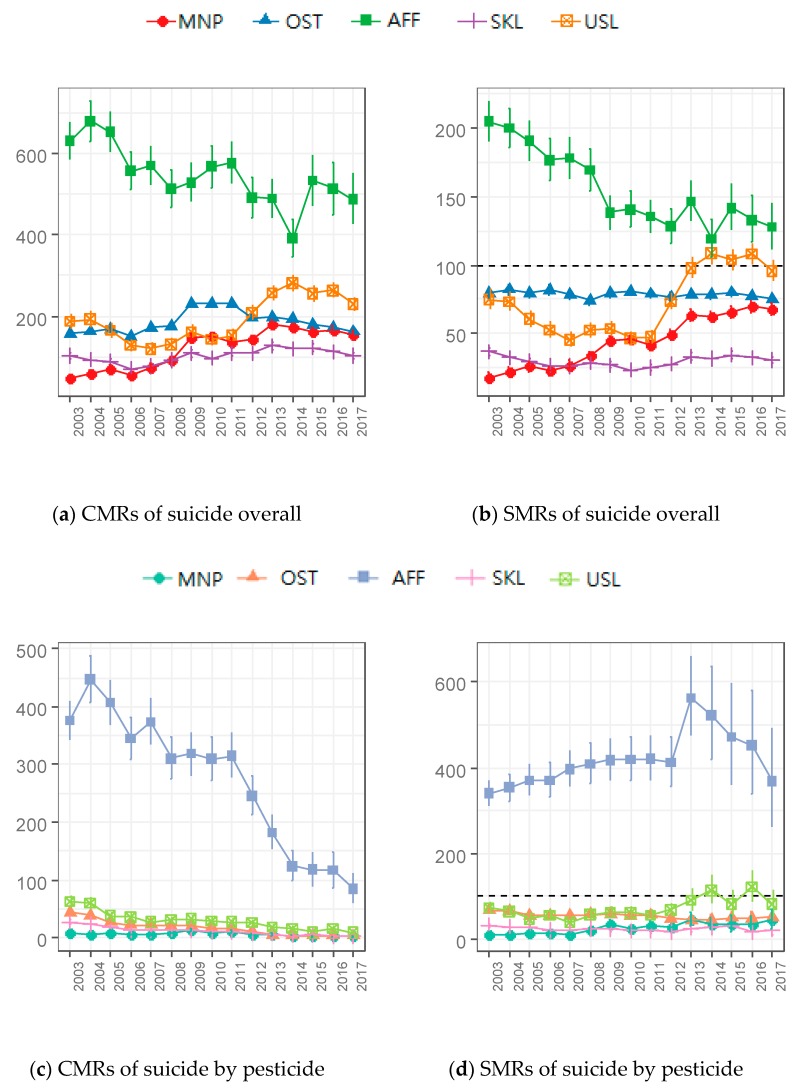
Time trends of suicide and suicide by pesticide rates by occupational groups among workers aged 15–64. The first row represents suicide overall and the second row represents suicide by pesticide. The left column shows the crude mortality rates (CMRs) and the right one shows the standardized mortality ratios (SMRs). MNP: Manager and professional, OST: Officer, service, and trade, AFF: Agriculture, forestry, and fishery, SKL: Skilled manual labor, USL: Unskilled manual labor. (**a**) CMRs of suicide overall. (**b**) SMRs of suicide overall. (**c**) CMRs of suicide by pesticide. (**d**) SMRs of suicide by pesticide.

**Figure 4 ijerph-16-04866-f004:**
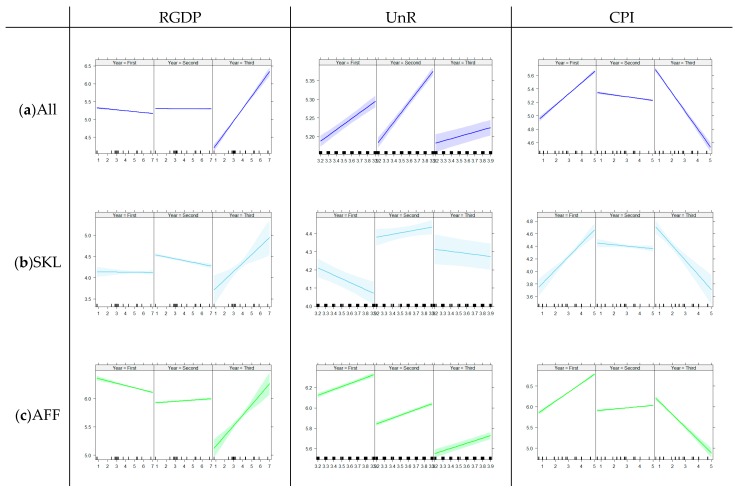
Summary of the general linear regression models for suicide by economic indicators. Each model shows each occupational group divided by each five-year period (2003–2007, 2008–2012, and 2013–2017). RGDP: real gross domestic product. UnR: unemployment rate. CPI: rate of customer price index. (**a**) Suicide overall rate for all occupational groups. (**b**) Suicide overall rate for the skillful manual labor group. (**c**) Suicide overall rate for agriculture, forestry, and fishery.

**Figure 5 ijerph-16-04866-f005:**
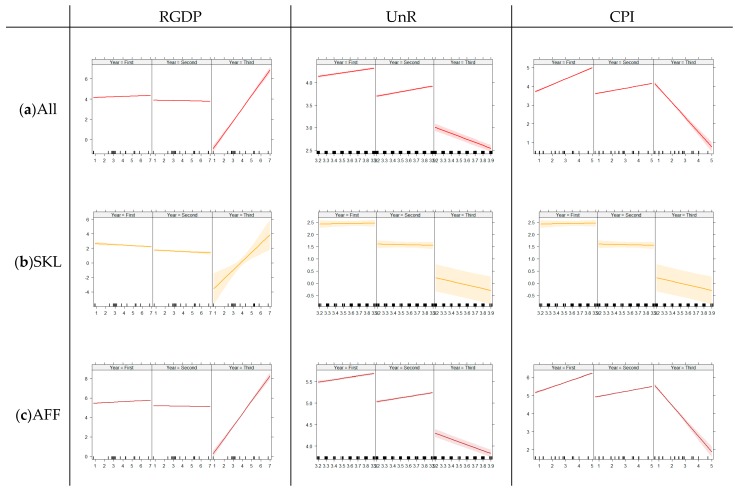
Summary of the general linear regression models for suicide by pesticide by economic indicators. Each model shows each occupational group divided by each five-year period (2003–2007, 2008–2012, and 2013–2017). RGDP: real gross domestic product. UnR: unemployment rate. CPI: rate of customer price index. (**a**) Suicide by pesticide for all occupational groups. (**b**) Suicide by pesticide for the skillful manual labor group. (**c**) Suicide by pesticide for agriculture, forestry, and fishery.

**Table 1 ijerph-16-04866-t001:** The standardized mortality ratios (SMRs) and crude mortality rates (CMRs) per 10,000,000 of suicide overall and suicide by pesticide by year among workers aged 15–64.

Year	SMR ^1^		CMR ^2^	
	Overall	Pesticide	Overall	Pesticide
2003	65.4 (63.2–67.7)	72.1 (68.0–76.4)	159 (153.9–164.8)	55 (52.5–59.0)
2004	63.7 (61.5–65.9)	70.3 (66.3–74.5)	158 (153.2–164.0)	54 (51.6–58.0)
2005	59.2 (57.1–61.2)	65.7 (61.4–70.1)	154 (149.5–160.2)	42 (39.3–44.9)
2006	55.9 (53.8–58.0)	63.0 (58.5–67.8)	128 (123.7–133.4)	33 (31.0–36.0)
2007	53.7 (51.8–55.7)	61.5 (57.0–66.2)	140 (135.2–145.3)	33 (30.5–35.4)
2008	55.3 (53.5–57.3)	63.7 (59.0–68.7)	147 (142.8–152.9)	30 (27.7–32.3)
2009	57.9 (56.1–59.6)	70.0 (64.8–75.4)	191 (185.3–196.9)	30 (28.5–33.1)
2010	56.2 (54.5–58.0)	63.7 (58.6–68.9)	186 (180.8–192.2)	26 (24.5–28.8)
2011	54.6 (52.9–56.2)	65.2 (60.1–70.6)	186 (181.0–192.3)	26 (24.4–28.6)
2012	58.8 (57.0–60.6)	62.2 (56.5–68.2)	177 (171.5–182.5)	19 (17.5–21.1)
2013	66.9 (65.0–68.9)	78.1 (69.6–87.2)	194 (189.2–200.6)	13 (11.7–14.7)
2014	66.0 (64.1–68.0)	74.2 (64.7–84.5)	189 (183.6–194.7)	9 (8.0–10.4)
2015	67.7 (65.6–69.7)	68.9 (58.8–79.9)	179 (173.9–184.8)	7 (6.0–8.2)
2016	67.1 (65.1–69.2)	67.5 (57.4–78.4)	175 (169.6–180.3)	6 (5.8–7.9)
2017	64.3 (62.2–66.3)	63.1 (52.6–74.6)	161 (156.5–166.8)	5 (4.4–6.3)

^1^ The ratio of the mortality of workers aged 15–64 to the general population aged 15–64, including workers and non-workers. The 95% confidence intervals are calculated by the Vandenbroucke method [30]. ^2^ The number of deaths determined assuming the population is 1,000,000.

**Table 2 ijerph-16-04866-t002:** Relative risks (RRs) for suicide and suicide by pesticide with age–sex stratification by occupational group to the skilled manual labor group among workers aged 15–64.

Strata ^1^	Predictors ^2^	Overall RR ^3^ (95% CI)	Pesticide RR ^4^ (95% CI)
All	(Intercept)	78.46 * (77.46–79.46)	7.76 * (7.45–8.08)
	MNP	1.44 * (1.42–1.47)	0.83 * (0.78–0.88)
	OST	2.44 * (2.40–2.47)	2.78 * (2.65–2.91)
	AFF	5.62 * (5.54–5.69)	25.49 * (24.46–26.57)
	USL	2.24 * (2.21–2.28)	3.08 * (2.94–3.23)
Younger Male	(Intercept)	85.85 * (83.77–87.97)	4.33 * (3.88–4.82)
	MNP	0.82 * (0.79–0.85)	0.52 * (0.43–0.63)
	OST	1.88 * (1.82–1.94)	2.35 * (2.07–2.68)
	AFF	5.89 * (5.74–6.05)	37.16 * (33.35–41.57)
	USL	2.05 * (1.99–2.11)	3.84 * (3.40–4.34)
Older Male	(Intercept)	136.79 * (134.16–139.45)	20.65 * (19.64–21.70)
	MNP	1.92 * (1.88–1.97)	0.88 * (0.82–0.94)
	OST	3.3 * (3.23–3.38)	3.01 * (2.84–3.19)
	AFF	5.69 * (5.58–5.82)	18.46 * (17.55–19.43)
	USL	3.17 * (3.10–3.24)	3.4 * (3.21–3.60)
Younger Female	(Intercept)	69.03 * (67.16–70.92)	2.95 * (2.58–3.35)
	MNP	0.65 * (0.62–0.68)	0.51 * (0.40–0.63)
	OST	1.26 * (1.22–1.31)	1.32 ^†^ (1.11–1.57)
	AFF	3.64 * (3.53–3.75)	39.73 * (34.86–45.53)
	USL	0.93 ^†^ (0.90–0.97)	1.13 (0.94–1.35)
Older Female	(Intercept)	22.16 * (21.11–23.24)	3.09 * (2.71–3.51)
	MNP	3.36 * (3.18–3.55)	1.21 ^†^ (1.01–1.44)
	OST	2.9 * (2.74–3.07)	3.22 * (2.78–3.73)
	AFF	10.23 * (9.73–10.76)	42.5 * (37.41–48.54)
	USL	1.33 * (1.25–1.42)	1.78 * (1.52–2.09)

^1^ Younger corresponds to 15–39 years old and older corresponds to 40–64 years old. ^2^ MNP: Manager and professional, OST: Officer, service, and trade, AFF: Agriculture, forestry, and fishery, USL: Unskilled manual labor. ^3^ Relative risks of the crude mortality rate for suicide. ^4^ Relative risks of the crude mortality rate for suicide by pesticide. * The *p*-value is under 0.001. ^†^ The *p*-value is under 0.05.

**Table 3 ijerph-16-04866-t003:** Relative risks by a simple logistic regression model using the crude mortality rate over each five-year period. The reference is the first five-year period, 2003–2007.

Jobs ^1^	Relative Risks for Suicide Overall		Relative Risks for Suicide by Pesticide	
	2008–2012	2013–2017	2008–2012	2013–2017
All	0.95 * (0.94–0.96)	1.01 ^†^ (1.00–1.02)	0.60 * (0.59–0.61)	0.22 * (0.22–0.23)
MNP	2.18 * (2.11–2.25)	2.66 * (2.58–2.74)	1.83 * (1.65–2.04)	0.69 * (0.60–0.79)
OST	1.15 * (1.13–1.17)	0.93 * (0.91–0.95)	0.53 * (0.50–0.56)	0.12 * (0.11–0.13)
AFF	0.69 * (0.68–0.70)	0.67 * (0.66–0.68)	0.59 * (0.58–0.60)	0.22 * (0.22–0.23)
SKL	1.22 * (1.18–1.26)	1.27 * (1.23–1.31)	0.49 * (0.44–0.53)	0.15 * (0.13–0.17)
USL	1.05 * (1.03–1.07)	1.64 * (1.61–1.67)	0.65 * (0.62–0.68)	0.29 * (0.27–0.31)

^1^ MNP: Manager and professional, OST: Officer, service, and trade, AFF: Agriculture, forestry, and fishery, SKL: Skilled manual labor, USL: Unskilled manual labor. * The *p*-value is under 0.001. ^†^ The *p*-value is under 0.05.

**Table 4 ijerph-16-04866-t004:** Linear regression models for suicide and suicide by pesticide with job stratification by the economic indicators for each five-year period.

Job ^1^	Year	Model for Suicide Overall ^2^				Model for Suicide by pesticide ^3^			
		β_0_	β_GDP_	β_UnR_	β_CPI_	β_0_	β_GDP_	β_UnR_	β_CPI_
All	2003–2007	4.38 *	−0.03 *	0.15 *	0.16 *	2.43 *	0.03 *	0.27 *	0.30 *
	2008–2012		−0.00	0.27 *	−0.03 *		−0.02 *	0.32 *	0.13 *
	2013–2017		0.34 *	0.06 *	−0.27 *		1.26 *	−0.67 *	−0.79 *
SKL	2003–2007	4.33 *	−0.00	−0.20 *	0.21 *	1.22 ^†^	−0.07 ^†^	0.06	0.52 *
	2008–2012		−0.04 *	0.08	−0.02		−0.06 ^†^	−0.07	0.34 *
	2013–2017		0.20 *	−0.06	−0.23 *		1.20 *	−0.74 *	−1.23 *
AFF	2003–2007	4.83 *	−0.04 *	0.28 *	0.22 *	3.78 *	0.05 *	0.28 *	0.25 *
	2008–2012		0.01 *	0.28 *	0.03 *		−0.01 *	0.30 *	0.14 *
	2013–2017		0.18 *	0.25 *	−0.31 *		1.29 *	−0.67 *	−0.85 *

^1^ SKL: Skilled manual labor, AFF: Agriculture, forestry, and fishery. ^2^ Generalized linear regression model for the crude mortality rate for suicide. ^3^ Generalized linear regression model for the crude mortality rate for suicide by pesticide. * The *p*-value is under 0.001. ^†^ The *p*-value is under 0.05.

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
