# Peer review of "Suicide Overall and Suicide by Pesticide Rates among South Korean Workers: A 15-Year Population-Based Study"

_ijerph, 2019, doi:10.3390/ijerph16234866_

Round 1

Reviewer 1 Report

Dear Authors! Thank you for a nice and interesting article, where you as most important findings shows that 1. workers with employment have a minor risk for comitting suicides compared to the population in general; 2. the use of pesticide for comitting suicide is decreasing, 3. while the overall number of suicides is not changing significantly. 4. One occupational group, farmers, forest workers and fishermen has a higher SMR than the other occupations and the general population andn 5. more often uses pesticides for suicidal acts. I only have a few suggestions for improvement: Your english sentences are not always easy to follow, so I suggest you consult a native english speaker for improvement of the language. In your Discussion part I miss reflections on the weaknesses of the study, this must be added. Best regards

Reviewer 2 Report

In this study authors aimed to analyze the association of  economical and  ecological factors, including the time before and after the pesticide ban policy was introduced and other with 15-year suicide rates. Results were stratified by age and sexTheir results show suicide by pesticide and overall suicide rates of workers in the agriculture, forestry, and fishery group are higher than other occupational groups and even the general population, and the effect of pesticide ban policy is also more significant.

This study is well done , the quality of presentation is high and wide-ranging . Minor points need to be taken into account before acceptation of publication. Please could authors enquire the following questions?

Is there difference in suicide rate between urban workers and rural workers? Could authors provide data on the homogeneity of the population and the source of heterogeneity Could authors provide data on confounding factors Could authors provide inclusion and exclusion factors Could authors compare the suicide rate between regions?

Reviewer 3 Report

This paper discusses an important subject within the field of suicidology. While the subject of suicide in South Korea and its association with various demographic variables has been discussed in other papers, this paper discusses the overall suicide rates and specifically looks at the association between employment in an agricultural industry and suicide by a particular method, which does not seem to have been done previously. Unfortunately, it was a challenging paper to get through because of both the language and the way the numbers were presented in the text. While the topic is worthwhile to discuss, because of the way the data are presented, I feel that the main objectives of the authors may be getting lost in a sea of numbers which are difficult for the general reader without a background in biostatistics to comprehend.

With regard to the study design, I have some concerns about the way that the authors have defined the populations according to age groups: while they have stated that they have chosen the age range of 15-64 as per the ILO definitions, I wonder if the population in South Korea has the same demographics when it comes to the working population – it would be reasonable to expect that in a more developed country, those at the younger end of this spectrum would still be students in high school or university, and that the age of retirement would be higher than 64. I think it would serve the authors better to define their working population based on local rather than international norms. I also wonder if it’s appropriate to define 15-39yo as “young” and 40-64 as “old”.

With regards to the presentation of the data, I have stated that the salient points seem to get lost in a sea of numbers and acronyms which form a large part of the text. Could the authors simplify the presentation of the information? Also, was there a reason for choosing to hyphenate some acronyms (eg A-F-F, M-P) and not others (eg SKL, USL)? A consistent rule for all the acronyms would make for easier reading.

There were many instances of awkward or unclear phrasing, and I have picked out some of these below. It is my opinion that less emotive and more objective language would serve the authors better in conveying their message convincingly. For example, the opening sentence, (lines 31-32) “South Korea has one of the highest suicide rates, and the situation has gotten worse in the past decades”—what do the authors mean by “highest”? Their next sentence seems to explain it, but the reader is required to make the inference themselves based on their own knowledge of global suicide statistics – do the authors mean that the suicide rate in South Korea is one of the highest in the world, and if so, perhaps they could state the global average or where South Korea falls in the ranking? And rather than stating that “the situation has gotten worse” (which reads like an opinion) perhaps they could state objectively that the rates have increased during the specified time period.

I would remove words such as “therefore” (line 33) and “moreover” (line 35) – do the authors know that the interventions were introduced because of the higher suicide rates in South Korea as compared to the rest of the world? Are the restrictions on the methods of suicide measures that were in addition to the “governmental and political interventions”? In line 35, the authors have paraphrased restricting access to lethal methods of suicide to “restrictions on the methods of suicide” but this seems to change the meaning; I think it would be reasonable to use the words in the title of the paper that is being referenced ([6]) without concern about plagiarism.

Line 37: “the Paraquat” sounds awkward. The sentence in lines 39-41 took me a moment to decipher, as I had to infer that “the complete ban” referred to a ban which was introduced in South Korea, rather than worldwide as (implied by the progression into talking about the WHO confirming the success of the policy).

Line 44: “older people are much riskier than younger” is again an awkward expression that doesn’t convey the meaning accurately; perhaps it should be rephrased. Line 47: “occupation affects suicide” could also be rephrased – the original sentence seems to mean that people with different occupations have different suicide rates, while the next sentence conveys a different meaning, that it’s employment status that has an association with suicide rates. Also, saying that “occupation affects suicide” implies a causal link, and it would be better to say that “there are associations between employment rates and type of occupation and suicide rates” (for example).

In line 49, what do the authors mean by “have worse conditions”? Do they mean that suicide rates are higher in these groups, or are they making an inference that people in these occupations are under worse living/working conditions than their peers? The meaning of the following sentence (line 49-50) is also not clear to me.

In the sentence in line 64-65, what do the authors mean by “the pesticide ban policy is also more significant”?

Line 154: it is not clear what is meant by “we can confirm the age and sex effect”

Line 159 – “inversed” is grammatically awkward, as is “the young become larger than the old”

Line 189: “enormous” is perhaps a subjective term; could the authors use more neutral language?

Line 192: what is meant by “have much more significant behaviour”?

Line 233: “affect” implies a causal link here

Line 260: “less risky for suicide” can be rephrased

Line 264-5: “the young are riskier than the old” can be rephrased, and “men are always more suicidal than women” sounds more like an opinion than a factual statement because of the phrasing.

Paragraph from line 261-266 -- could the authors offer some theories on why these trends might exist?

Line 268-9: how does the present study confirm this? What is meant by "a good effect on suicide"?

Line 285-6: the meaning of this sentence is unclear; reductions in what? Employment status or suicide rates?

Line 288-9: This sentence is again unclear. What is meant by “more reductions in the general population of workers”? Do the authors mean reductions in the proportion of the population which is employed, or that the population of workers decreased because of death?

Line 327: spelling error “county” instead of “country”

Line 357-8: “had shown risky conditions concerning suicidality which were even much worse” is grammatically awkward

Line 359-360: suicide by pesticide decreased needs to be qualified – decreased over a time period, or decreased after some intervention etc

Tables and figures:

Table 5 header – is there a typo in “Year1”?

For all the figures, perhaps a key could be given specifying what each shape on the graph represents in addition to colour-coding the graphs, as it is not possible to tell the difference once the paper is printed out in black and white?
